# Towards the Elaboration of a Non-Technical Skills Development Model for Midwives in Morocco

**DOI:** 10.3390/healthcare10091683

**Published:** 2022-09-03

**Authors:** Asmaa Ghafili, Abdellah Gantare, Claire Lobet-Maris, Maximilien Gourdin

**Affiliations:** 1Nursing and Midwifery Unit, Laboratory of Health Sciences and Technologies, Higher Institute of Health Sciences, Hassan First University of Settat, Settat 26000, Morocco; 2Faculty of Public Health, Université Catholique de Louvain, 1200 Woluwe-Saint-Lambert, Belgium; 3Royal Academy of Science, Letters and Fine Arts of Belgium, BE1000 Bruxelles, Belgium; 4Technology, Ethics & Society Research Unit, Research Centre in Information, Law and Society (CRIDS), Namur Digital Institute (NADI), Faculty of Computer Sciences, University of Namur, 5000 Namur, Belgium; 5Department of Anesthesiology, UCLouvain, CHU UCL Namur, 5530 Namur, Belgium

**Keywords:** assessment, education, midwives/midwifery, management, non-technical skills

## Abstract

This article explores the non-technical skills critical for the practice of midwives through a comparison of two maternity services in Morocco. Soft skills, or non-technical skills, present a set of metacognitive abilities, which complement hard or technical skills, in order to guarantee the safe performance of a technical activity. This exploration is based on an original methodology that triangulates observation of caring paths, qualitative interviews, and quantitative questionnaires. We identified the main soft skills mastered, those that were missing, and those to be developed, based on an observed or expressed need. The research population included 30 midwives and 70 women. The results led us to identify the most critical non-technical skills for midwifery practice at a Local Medical Centre (LMC) and a Provincial Hospital Centre (PHC) to better understand the effects of workload on the possibilities of activating non-technical skills during caring paths. Based on these results, we elaborated a model for the development and improvement of non-technical skills in midwifery.

## 1. Introduction

In this article, we focus on soft skills as they are related to the work of midwives. Soft skills or non-technical skills (NTS) have been described in the literature as a combination of interpersonal and social skills that complement technical skills and contribute to safe and effective technical performance [1]. They referred to all skills that are not directly related to a specific task but to interactions with other people in an organisation; they are necessary for any position and are therefore transferrable. Technical skills, on the contrary, refer to the specific skills needed to perform a particular task [2]. Technical skills are conceived as a type of easily documented and formalised knowledge, which could produce something visible and direct [3].

The soft skills, or non-technical skills present a set of metacognitive abilities, which are complemented with the hard or technical skills, in order to guarantee the achievement of a safe technical activity and an efficient service [4]. In the medical field, for example, the majority of errors with serious consequences for patients are due to a failure of medical staff, specifically in fields such as surgery, anaesthesia, or even emergencies.

Our analysis focuses on midwives, who play a leading role in the effective management of childbirth in Morocco. The literature has showed that it is essential for midwives to develop their adaptability, critical and strategic thinking, emotional intelligence, and leadership skills throughout their careers [5].

Utami et. al (2017) [6] considered that “the more non-technical skills that a person possesses, it is expected that they will have a stronger personality when facing the challenges of the learning process, work challenges, and other life challenges… a strong attitude, are honest, passionate, able to work together, polite in communicating, good at negotiating, have a high work motivation, are creative and adaptable, and so are able to work intensively”. They developed a model “Leadership and communication skill model on midwifery students in physiological delivery practice”, based on eight non-technical skills: motivation, self-leadership, job satisfaction, psychological, empowerment, self-efficacy, task commitment, leadership skills, and communication skills.

In order to describe and understand the relationship between the soft skills involved in the midwife/parturient relationship and quality of care, the objective of this research was to explore the NTS critical to the practice of midwives through a comparison of two services in health institutions in the Casablanca-Settat region: the first one in the Provincial Hospital Centre (PHC) and the second one in the Local Medical Centre (LMC). This choice was justified by the contrast between them in terms of hospitalisation rate, number of midwives, and team workload. To meet this objective, we developed an original methodology to assess the NTS mobilised by midwives in their caring practices. The ambition of the research was also that this methodology tested at two sites could give rise to the development of a self-evaluation tool of NTS for midwifery teams.

In our study we conducted interviews with midwives and women in labour, distributed self-assessment questionnaires to the midwives, and assessments to the women, and then observed care journeys. The confrontation of the results allowed us to elaborate a model containing the most critical non-technical competences in the midwifery profession.

## 2. Materials and Methods

The empirical framework was based on the triangulation of three methodologies: observation of the care pathway, qualitative interviews, and a quantitative questionnaire. Inspired by the research of Luis et al. (2015), Piyawan et al. (2015), Bracco et al. (2017), Engels (2017), and El filali (2021) [7,8,9,10,11], we designed our data collection framework around four main dimensions of NTS, namely psychological, organisational, cognitive, and interactional.

(1) The first methodological approach was based on the observation of the NTS in situations as mobilised during caring paths. The structure of a typical caring path was designed, and a framework of observation was developed based on the four dimensions of NTS mentioned above. Seven caring pathways were monitored and analysed: two in the LMC and five in the PHC.

(2) The second stage of data collection consisted of semi-structured interviews with women and midwives. The objective of these interviews was to explore in more detail the NTS related to the four dimensions mentioned above, and to compare the points of view of the two populations on the NTS mobilised during care. We conducted a total of 30 interviews with midwives, including 10 in LMC and 20 in PHC. For women, we went to the postpartum service, where we conducted 70 interviews, namely 20 interviews at the LMC and 50 at the PHC.

(3) The third step was based on a quantitative questionnaire to assess and compare the level of satisfaction of the midwives and the women with respect to NTS mastered and mobilised during care. The population considered by the questionnaire was similar to the one in the second step (70 women and 30 midwives). To elaborate this questionnaire, we synthesized the results of the qualitative interviews and the consulted literature. This allowed us to refine each of the four dimensions mentioned above into five key competencies, to be assessed by the two populations.

## 3. Results

In this section, we will present the results according to the three stages of our methodological triangulation. Before that, we give a brief overview of the two sites at which the research took place.

### 3.1. The Two Research Sites: Contrasting Realities

Operating indicators, collected at the two sites, demonstrated a significant contrast in the workload of these two populations of midwives. The LMC had a total of 12 midwives, of which 2 to 3 midwives formed a team based on a 12 h/36 h system. The team’s average seniority was 10. In 2021, the number of births carried out was 720, which meant a monthly average of 5 births by midwives. At the PHC, there were 28 midwives, of which 3 midwives formed a team based on a 12h/36h system. The average seniority was 15 years. Each PHC midwife performed almost 25 births on average per month, for a total of 7639 births in 2021.

### 3.2. Situational Observations: The Caring Path

A typical caring path was designed and structured in different stages (Figure 1).

We made our observations of the attitudes, behaviours, and skills involved in each step of the caring path in Figure 1 that the women followed in order to benefit from gyno-obstetric care. We observed the complementarity of non-technical skills with technical skills in each step of the caring path: the anamnesis (communication) at the same time as the clinical examination, and an explanation of the gestures to relieve pain at the moment of giving birth. We pointed out that some elements of communication skills remained poorly mobilised during the observed caring paths in both teams, for instance, the personalised interaction with the woman in labour. We also found that the mobilisation of some soft skills tended to decrease in critical situations. This was the case for skills relating to the team’s organisation and coordination, calm communication with the woman, the involvement of the woman and her partner in both care and decision-making, and finally skills related to the respect for intimacy.

### 3.3. Qualitative Assessment: Interview Results

The interviews carried out with each of the midwives had four axes: quality of services, the performance of the service, mastery of NTS, and strategy for their development. The interviews lasted an average of 15 min with each midwife.

Interviews with the women focused on their assessment of the quality of the attitudes of midwives during care, attitudes considered unsatisfactory, and attitudes they expected in their interactions with midwives. Each interview lasted 20 min on average.

After translating the verbatim transcripts from the Arabic language to the French language, we used the ATLAS-Ti software to analyse the data collected. This allowed us to refine the conceptualisation of the categories to enrich our four initial dimensions, and to give them content through a more precise identification of the NTS.

#### 3.3.1. Results of Interviews with Midwives

Regarding the criteria related to the quality and the performance of the service, at the LMC, 40% pointed out the qualification of human resources and the women’s satisfaction as evaluation criteria, while at the PHC, only 20% mentioned these criteria. In both units, the importance of non-technical skills for the quality and the performance of the service was validated, at 85% for the PHC, and 90% for the LMC. Regarding the NTS mastered (Table 1) by the respondents, the results showed no difference between the two sites. Considering the missing non-technical skills at the level of the department, we noticed a difference between the two sites: midwives at the LMC noted their marginalisation in organisational decision-making, while the midwives at the PHC expressed that several skills were lacking in their department, such as teamwork, conflict management (as a group), and acceptance of criticism, which should be developed and strengthened at the service level. These differences in results could be explained by the contrasting workload experienced by the two teams. Regarding the last question related to their proposals for the development of the missing NTS, all the midwives at the two sites pointed to continuing education as the best way to improve them. However, there was a strong difference between the two sites regarding the conditions for this improvement. In the less loaded environment (LMC), midwives pointed out personal motivation, whereas in the more demanding work environment, midwives emphasised structural conditions relating to human resources and the working climate.

#### 3.3.2. Results of Interviews with Women

At the level of the LMC, all the women expressed their satisfaction with the quality of the caring attitudes during the examination by the midwives, while at the level of the PHC, 50% of the women were dissatisfied.

Women at the LMC level pointed out the mastery of some non-technical skills, for example, composure (the midwife was able to face situations without adopting a defensive or emotional attitude), active listening by responding to needs and requests, and adaptability. On the other hand, they expressed an intense need for their involvement in the care, in order to present their points of view. Moreover, they pointed out that the skills of pain relief, communication, and good decision-making are usually the most important to them and therefore the most expected from midwives. At the level of the PHC, concerning the mastered skills, women emphasised the quality of the practice of care and the response to their needs. Regarding the missing NTS, they mentioned several: communication, conflict management, respect for intimacy, and teamwork. They explained their dissatisfaction with inappropriate body language and lack of communication experienced during the care. In terms of the most expected skills, they focused on pain management, communication, and active listening.

Table 2 summarises the non-technical skills (NTS) mastered by the caregivers (midwives) according to the women, as well as those missing and those expected during their caring journey.

### 3.4. Quantitative Level of Satisfaction: Results of Questionnaires

The aim of the questionnaire was to assess and compare the level of satisfaction (on a scale of 5:1 for excellent and 5 for unsatisfactory) of midwives and women with respect to NTS mastered and mobilised during care. We distributed 100 questionnaires to 70 women (50 in PHC, 20 in LMC), and 30 midwives (20 in PHC, and 10 in LMC), all of which were collected, and used in the study. By comparing the two populations (women and midwives) in the Table 3, it was interesting to note that the assessments of women were slightly behind those of midwives, signifying a gap between the perceptions of midwives and women. This difference in perception was more strongly marked in the PHC where, for each NTS dimension, more than 30% of the women assessed it as to be improved (4) or unsatisfactory (5). The comparison of the self-assessment of the midwives at the PHC and the LMC showed clear differences in their level of satisfaction. For each dimension, the level of satisfaction (1–2) was higher in the LMC than in the PHC. These findings suggested a clear influence of workload on the potential for NTS mobilisation in interactions and care.

## 4. Discussion

In our research, we used three data collection devices: observation of caring paths, qualitative assessment based on interviews, and questionnaires of satisfaction. We collected those data from both the women and the midwives. This interconnection between methods and between populations allowed us to better understand the issue of NTS for midwifery.

The observations on the two fields facilitated the detection of organisational and interactional NTS critical for the quality of care. The qualitative self-assessment helped us to enrich our four initial dimensions by giving them empirical realities and contents, but also by pointing out psychological and cognitive NTS that our study population (the midwives and the women) deemed critical and/or as NTS that needed be improved in the care. The questionnaire was also critical in validating the differences in perceptions between the women and the midwives, but also in identifying the importance of the workload in NTS mobilisation opportunities.

In particular, our quantitative results showed a clear difference between:
What is thought to be mastered by the midwives’ staff, and what was expected by the women. These differences clearly demonstrated the need to develop a model to support the training of midwives in NTS.The results from the two sites demonstrated that it was necessary to approach the NTS in a global context, namely that of the workload. Indeed, we pointed out that in the context of stress and heavy workload at the PHC level, the majority of NTS did not have enough opportunities to be deployed. This result demonstrated the need to consider the NTS development in a global approach, along with the human resources and managerial conditions.

The results of our three methodological tools helped us to identify the most critical NTS in demand in midwifery practice for both sites. Based on them, we elaborated a model for developing and improving the NTS of midwives (Figure 2). We named this model POCI, taking each first letter of its four constituent dimensions.

Regarding the psychological dimension (P), our results pointed out two key skills that each team member should possess. The first was the ability to remain motivated and engaged despite the stress of the workload. This was a skill especially highlighted by PHC midwives. The second, very much related to the first, was their ability to manage the ongoing stress of their workload and of critical care situations. This ability to manage stress and critical situations was conditioned by the mobilisation of two critical organisational skills (O): managing conflicts and effectively coordinating the team’s work, as well as that of other stakeholders such as gynaecologists or anaesthetists. For the cognitive dimension (C), our results showed the importance of timely clinical problem-solving, either individually or in a team setting. The results also pointed out the need for significant agility and adaptability in dealing with the variety of clinical situations faced by the midwives. Lastly, concerning the interaction (I), the ability to communicate calmly and clearly, whether among themselves or with the women, was considered by our population to be critical to ensuring safe care and patient well-being. Last but not least, the well-being of women also depended on being shown respect, which called for the ability of midwives to protect intimacy while caring for them.

Our study showed the need to include the development of midwives’ non-technical skills in their continuing education, since they did not benefit from it in university programs. On that same point, a study conducted with nursing students in a non-technical skills training program found that it was an effective method of improving emotional intelligence and resilience, and improving the nursing students’ personal and professional competencies, resulting in higher quality care. As a result, it was recommended as a psychoeducational strategy for training undergraduate nursing students [12].

In order to validate the main soft skills inherent in our POCI model, we compared them with those of the synthetic model of IDEAc soft skills [13]. This model is made up of 10 soft skills, which are critical for success in the professional world, grouped into five areas of competence. The five skill areas are influence (leadership and self-awareness), decision making (critical thinking and entrepreneurial spirit), efficiency (learning and solving complex problems), agility (creativity and mental flexibility), and cooperation (empathy and collaboration). By analysing these five areas, it appears that the authors’ categories can be integrated into our organizational dimension: decision making and agility is the organizational dimension, efficiency is part of the cognitive dimension, and finally the field of cooperation is part of the psychological and organizational dimension.

Our POCI model is more synthetic and thus meets the scientific criterion of parsimony.

Our modelling finds favourable echoes in various studies [14,15], especially since they include most of the certified soft skills (stress management, communication, teamwork, problem solving, and adaptation to complex situations, among others). However, our model has the merit of giving solid foundations to the different soft skills by linking them to four major dimensions.

In addition, it was built in an anchored and empirical way through the study of two very contrasting cases regarding the workload in the care units. This empirical construction also made it possible to approach the systemic character of the model, with the four dimensions being closely interconnected.

Nevertheless, it is a perfectible model which must be tested with a team of midwives within the ForSim simulation centre of our institute in order to refine it and see to what extent it is transferable.

## 5. Conclusions

In conclusion, the study showed us that our methodological protocol, tested at two sites and based on the triangulation of methods, could help departments to set up a self-assessment process for the critical NTS mobilised in order to strengthen the quality and the safety of their care. Our study gained a better understanding of the effects of workload, as approached by our operational indicators, on the possibilities of activating NTS during the caring path, as well as on the types of NTS that tend to fade in critical situations or related to the stress caused by an excessive workload. This understanding invited us to consider possible improvement to NTS in a global approach that integrates human resource management and work organisation. Moreover, our POCI model was made up of eight non-technical skills, which were critical for success in the professional world, grouped into four dimensions of competence.


**Limitations:**


The POCI model has been developed from two different contexts, and will specifically address real needs, hence the importance of applying our methodology in other services in order to adjust the critical skills to their own POCI model.The relationship between midwives and other staff was not included in our study.


**Future lines of research:**


As future lines of research, we currently intend to test the empirical validity of our modelling by mobilizing simulation both at the level of initial and continuing training of midwives in order to develop the various soft skills identified primarily at the field level of our research.

## Figures and Tables

**Figure 1 healthcare-10-01683-f001:**
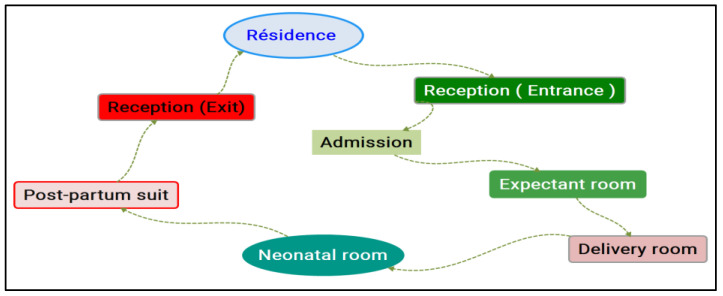
Treatment path followed during the observations.

**Figure 2 healthcare-10-01683-f002:**
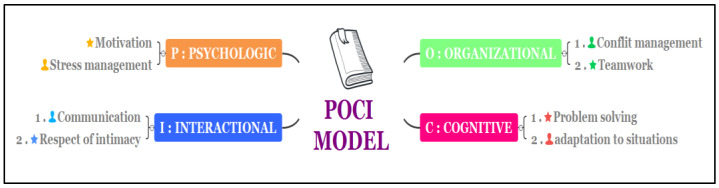
The essential non-technical skills according to the dimensions of POCI skills.

**Table 1 healthcare-10-01683-t001:** Mastered and missing NTS and their development strategy according to the midwives.

	LMC	PHC
**NTS mastered by the respondents**	Stress management, decision-making, conflict management, communication	Stress management, decision-making, conflict management, communication
**NTS missing in the department**	Involvement in organisational decision-making	Teamwork, conflict management, acceptance of criticism
**Development of NTS**	Continuous training, motivation.	Continuous training, availability of human resources, working climate

**Table 2 healthcare-10-01683-t002:** Mastered, missing, and expected NTS from the point of view of the women.

	LMC	PHC
**-NTS mastered (from their actual experience)**	Self-control, listening, adaptability to the situation	The practice of childbirth, care of newborns, response to needs
**-NTS missing (during their experience)**	Involvement in care	Communication, conflict management, respect for intimacy, teamwork, initiative
**-Expected NTS (by the women at the time of treatment before admission to the service)**	Pain relief, communication, good decision-making	Pain relief, active listening, communication, problem-solving.

**Table 3 healthcare-10-01683-t003:** Level of satisfaction of midwives and women with respect to NTS mastered and mobilised during care.

Dimensions:		Psychological %					Organisational %					Cognitive %					Interactional %				
**Rating Levels:**		**1**	**2**	**3**	**4**	**5**	**1**	**2**	**3**	**4**	**5**	**1**	**2**	**3**	**4**	**5**	**1**	**2**	**3**	**4**	**5**
**PHC**	**Midwives n = 20**	**0**	**8.7**	**55**	**36.2**	**0**	**2.5**	**26.6**	**49.1**	**21.6**	**0**	**0**	**27**	**52**	**21**	**0**	**12.5**	**17.5**	**30**	**40**	**0**
	**Women** **n = 50**	**0**	**23**	**39**	**22**	**16**	**0**	**13**	**51.6**	**18.3**	**17**	**0**	**15.6**	**52, 8**	**14.4**	**17.2**	**8.3**	**25**	**28**	**17.3**	**21.3**
**LMC**	**Midwives** **n = 10**	**0**	**40**	**40**	**20**	**0**	**15,**	**55**	**11.6**	**13, 3**	**5**	**20**	**32**	**22**	**18**	**8**	**26.6**	**36, 6**	**16.6**	**15**	**5**
	**Women** **n = 20**	**0**	**48.7**	**51.2**	**0**	**0**	**0**	**28.3**	**71.6**	**0**	**0**	**0**	**30**	**70**	**0**	**0**	**14.1**	**54.1**	**31.6**	**0**	**0**

## Data Availability

Data are available from the corresponding author upon reasonable request.

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
