# Peer review of "Towards the Elaboration of a Non-Technical Skills Development Model for Midwives in Morocco"

_healthcare, 2022, doi:10.3390/healthcare10091683_

Round 1

Reviewer 1 Report

Very interesting topic, the researcher did a great job to

 explore the non-technical skills critical to the practice of midwives through a comparison of two maternity services. However, the paper could be strengthened through:

Add more towards scope of the problem in introduction section.

Add current references to strengthen the paper.

Good luck

Author Response

Dear Reviewer,

We would like to thank you for your comments, they were very useful for improving the quality of the paper. We have thoroughly revised the manuscript, in the attachement file , we will explain in detail how we addressed all of your comments.

Thank you.

Reviewer 2 Report

Dear Authors

I find the article presented to me interesting and necessary.

However, it needs some corrections.

I believe that the introduction requires an expansion and a more detailed description of the theoretical foundations of the research.

In my opinion, the conclusions also need to be corrected. They are very complex and should be a synthesis.

I have no objections to the rest of the work.

Yours sincerely, Reviewer

Author Response

(The authors gave the same response as above.)

Reviewer 3 Report

Thank you for giving me the opportunity to review the article entitled "Towards The Elaboration of a Non-Technical Skills Development Model for Midwives in Morocco".

Although this is a very interesting topic, I believe that the article requires some improvements in order to be published:

In the introduction, the authors could give some practical examples of

 soft skills and  non-technical skills. In addition, I consider the introduction to be incomplete. The authors should include aspects such as the GAP to be covered by this research, main contributions, research questions to be answered and a final paragraph showing the structure of the remaining article.

It would be interesting if the authors worked a little more on the state of the art.

In the discussion section, the results are not compared with those obtained by other researchers. On the other hand, the conclusions lack contributions, practical applications, limitations to the scope and future lines of research.

Author Response

(The authors gave the same response as above.)

Round 2

Reviewer 3 Report

I am very pleased to note that the authors have followed all my recommendations. Therefore, the article can now be published.